# Investigating *Mycobacterium tuberculosis sufR* (*rv1460*) *in vitro* and *ex vivo* expression and immunogenicity

**Lucinda Baatjies**[1]*, **Ilana C. van Rensberg**[1], **Candice Snyders**[1], **Andrea Gutschmidt**[1], **Andre G. Loxton**[1☯], **Monique J. Williams**[2☯]

**1** Division of Molecular Biology and Human Genetics, Department of Science and Innovation (DSI)-National Research Foundation (NRF) Centre of Excellence for Biomedical Tuberculosis Research, South African Medical Research Council Centre for Tuberculosis Research, Faculty of Medicine and Health Sciences, Stellenbosch University, Cape Town, South Africa, **2** Department of Molecular and Cell Biology, University of Cape Town, Cape Town, South Africa

☯ These authors contributed equally to this work.
* lbaatjies@sun.ac.za

**Data Availability Statement:** All relevant data are within the paper and its Supporting Information files.

## Abstract

Iron is vital metal for *Mycobacterium tuberculosis* infection, survival, and persistence within its human host. The mobilization of sulphur (SUF) operon encodes the primary iron-sulphur (Fe-S) biogenesis system in *M. tuberculosis* and is induced during iron limitation and intra-cellular growth of *M. tuberculosis*, pointing to its importance during infection. To study *sufR* expression at single cell level during intracellular growth of *M. tuberculosis* a fluorescent reporter was generated by cloning a 123 bp *sufR* promoter region upstream of a promotor-less *mcherry* gene in an integrating vector. Expression analysis and fluorescence measurements during *in vitro* culture revealed that the reporter was useful for measuring induction of the promoter but was unable to detect subsequent repression due to the stability of mCherry. During intracellular growth in THP-1 macrophages, increased fluorescence was observed in the strain harbouring the reporter relative to the control strain, however this induction was only observed in a small sub-set of the population. Since SufR levels are predicted to be elevated during infection we hypothesize that it is immunogenic and may induce an immune response in *M. tuberculosis* infected individuals. The immune response elicited by SufR for both whole blood assay (WBA, a short term 12-hr stimulation to characterise the production of cytokines/growth factors suggestive of an effector response) and lymphocyte proliferation assay (LPA, a longer term 7-day stimulation to see if SufR induces a memory type immune response) were low and did not show a strong immune response for the selected Luminex analytes (MCP-1, RANTES, IL-1b, IL-8, MIP-1b, IFN-g, IL-6 and MMP-9) measured in three clinical groups, namely active TB, QuantiFERON positive (QFN pos) and QFN negative (QFN neg) individuals.

**Funding:** This work was supported by the South African Medical Research Council (SAMRC), DST-NRF Centre of Excellence for Biomedical Tuberculosis Research (CBTBR). AGL is supported by the EDCTP TESA network (CSA2020NoE-3104). MW was supported by a research career award from the National Research Foundation (Grant number: 91424). The funders had no role in study design, data collection and analysis, decision to publish, or preparation of the manuscript.

**Competing interests:** The authors have declared that no competing interests exist.

**Abbreviations:** CD, cluster differentiation; CFU, colony forming units; Fe-S, iron-sulphur; hMDMs, human monocyte-derived macrophages; HREC, Human Research Ethics Committee; hrs, hours; hyg, Hygromycin B; IFN-g, interferon gamma; IL-1b, Interleukin -1beta; IL-6, Interleukin-6; IL-8, Interleukin-8; LB, Luria-Bertani; LPA, lymphocyte proliferation assay; LTBI, latent TB infection; MCP-1, monocyte chemoattractant protein-1; MFI, median fluorescent intensity; min, minutes; MIP-1b, macrophage inflammatory protein-1beta; MMP-9, matrix metallopeptidase-9; MOI, multiplicity of infection; M. tuberculosis, *Mycobacterium tuberculosis*; NaHep, sodium heparin; OADC, oleic albumin dextrose catalase; OD, optical density; QFN neg, QuantiFERON negative; QFN pos, QuantiFERON positive; PHA, Phytohaemagglutinin; RANTES, regulated and normal T cell expressed and secreted; RT, reverse transcription; s, second; SUF, mobilization of sulphur; TB, Tuberculosis; WBA, whole blood assay; WHO, World Health Organization; wt, wild type.

## Introduction

A significant event in the history of medicine and in our understanding of the deadly disease tuberculosis (TB), was the discovery of the tubercle bacillus by Robert Koch in 1882 [1]. Despite this knowledge, TB has killed 1.2 billion people over the last 200 years and was only temporarily displaced by COVID-19 as the leading cause of death due to an infectious agent in 2020 and 2021 [2–4]. The 2022 World Health Organization (WHO) TB report showed that progress towards TB milestones and targets has been hard hit by the COVID-19 pandemic. Fewer people were diagnosed and treated for TB, with newly diagnosed cases decreasing from 7.1 million in 2019 to 5.8 million in 2020 [4]. Furthermore, an increase in the number of TB-related deaths was observed between 2019–2021, increasing to 1.6 million from an estimate of 1.5 million in 2020 and 1.4 million in 2019 [4].

TB remains a global problem partly because current standard diagnostic strategies and tools are inadequate, resulting in delayed initiation of treatment. This has serious outcomes for global TB control efforts [5], as early diagnosis and treatment reduces the risk of transmission [6]. The shortcomings of current immunodiagnostics include the inability to detect progression from latent TB infection (LTBI) to active TB, and the inability to monitor treatment [7]. Several studies have therefore sought to identify new biomarkers for diagnosis and treatment response [8]. One strategy for identifying proteins with diagnostic potential is to identify pathways required by the causative agent of TB, *Mycobacterium tuberculosis*, (*M. tuberculosis*) during infection [7]. Iron is essential micronutrient required by most pathogens to survive in their human host [9, 10] and the importance of iron for the pathogenesis of *M. tuberculosis* has been demonstrated by several studies [11–13]. *M. tuberculosis* is hypothesized to experience iron-limitation within the host [10, 14] and one of the processes upregulated in response to iron-limitation is iron-sulphur (Fe-S) cluster biosynthesis; Fe-S cluster biogenesis therefore seems to be prioritized by *M. tuberculosis* when iron is limiting to ensure that essential Fe-S cluster proteins remain functional [15].

The *M. tuberculosis* SUF system is the primary Fe-S biogenesis machinery in *M. tuberculosis* [16] and is encoded by a single operon (*sufR-sufB-sufD-sufC-csd-nifU-sufT*). Apart from *sufR*, which encodes a transcriptional repressor of the operon, all the genes in the operon are predicted to be essential for viability [17–19]. The importance of SufR during infection is supported by i) induction of *sufR* in *M. tuberculosis* in sputum from TB patients relative to expression *in vitro* [20] and ii) detection of anti-SufR antibodies in the serum of TB patients [20]. Despite the importance of iron homeostasis during TB infection, the utility of *M. tuberculosis* proteins induced by iron limitation in the host as diagnostic antigens remains largely unexplored. This study therefore aimed to investigate *sufR* expression during intracellular growth and to investigate the diagnostic potential of the SufR protein.

## Materials and methods

### Bacterial strains growth and culture conditions

*M. tuberculosis* H37Rv, H37Rv_*attB*::pMV306 (wild type (wt) control) and H37Rv_*attB*::pMV306_123mCherry (S1 Table in S1 File) were cultured in Middelbrook 7H9 broth (Difco, BD) supplemented with 0.05% Tween 80, 0.2% glycerol and Middelbrook oleic albumin dextrose catalase supplement (BD) (7H9 OADC) or on 7H10 (Difco, BD) supplemented with 0.5% glycerol and Middelbrook OADC (7H10 OADC). Hygromycin B (hyg) (50 μg/mL) (Inqaba Biotech) was used for selection purposes. Starter cultures were incubated at 37°C for 5–6 days and then sub-cultured to an optical density 600 nm ($OD_{600nm}$) of 0.05. Cultures were incubated at 37°C and growth monitored by optical $OD_{600nm}$ or determined by enumerating

colony forming units (CFUs) plated on 7H10 OADC. The *Mycobacterium smegmatis* mc$^2$155 strain containing pCHARGE3 (S1 Table in S1 File) was cultured overnight in 7H9 OADC (Difco, BD) at 37˚C, shaking at 200 rpm (Yihdern LM 530 orbital shaking incubator, Yihder, Taiwan). The *E. coli* XL1 Blue strain (S1 Table in S1 File) was cultured in Luria-Bertani (LB) broth at 37˚C shaking at 200 rpm (Yihdern LM 530 orbital shaking incubator, Yihder, Taiwan). Rubidium chloride competent cells were prepared using this *E. coli* strain as previously described [21].

## Construction of *Mycobacterium tuberculosis* fluorescent reporter strains

All PCR reactions were performed with using the Phusion High-Fidelity PCR Kit (Thermoscientific, Massachusetts, USA) according to the manufacturer's instructions. The *sufR* promotor (123 bp) was amplified from H37Rv genomic DNA using primers SufRPf and SufRPr (S2 Table in S1 File). The promoterless *mCherry* fluorescent gene was amplified from pCherry3 plasmid (Addgene plasmid# 24659; http://n2t.net/addgene:24659; RRID: Addgene_24659) with a forward primer (Ovlp) and reverse primer mCherR (S2 Table in S1 File). An overlap PCR was then performed to fuse the *sufR* promoter to the promoterless *mCherry* fragment using primers SufRPf and mCherR (S2 Table in S1 File). The fused PCR product (sufR_123mCherry) was cloned into the sequencing vector pJET1.2 (S1 Table in S1 File) and following confirmation by Sanger sequencing, was subcloned into the HindIII and EcoRV sites in the pMV306 plasmid (S1 Table in S1 File).

## RNA extraction, cDNA synthesis and qPCR

For total RNA extraction early-log (day 4), mid-log (day 10) and early stationary phase (day 14) cultures (5 ml) were pelleted, resuspended in 1ml RNAProBlue solution (Iepsa) and ribolysed for three 30 seconds (s) cycles at 4.5 watt using the FastPrep-24 Instrument (MP Biomedicals). Cellular debris was removed by centrifugation at 16 000 x g for 15 minutes (min) at 4˚C. Chloroform extraction was performed, and RNA precipitated with ethanol prior to loading onto a Nucleospin RNA kit (MacheryNagel) column. An on-column rDNase treatment were performed according to the manufacturer's instructions and RNA eluted in 30 μl nuclease free water. RNA quality and concentration were assessed using the Agilent Bioanalyzer RNA Nano Assay kit. RNA was DNase treated using the TURBO DNA-free kit (Ambion), RNA (100 ng) was reverse transcribed using the Transcriptor first strand cDNA synthesis kit (Roche) according to manufacturer's instruction using 1 μM of reverse transcription (RT) primers indicated in S2 Table in S1 File. RT-qPCR was performed with the FastStart Essential DNA Green Master mix using 0.1 μM PCR primers as indicated in S2 Table in S1 File and the LightCycler 96 machine (Roche). A standard curve was generated for each primer set using genomic DNA dilutions, and expression levels calculated relative to the reference gene, *sigA* [19].

## THP-1 infections

All incubations were done at 37˚C in a humidified incubator (ESCO Vivid Air) with 5% $CO_2$. THP-1 cells were cultured in RPMI 1640 with L-Gutamine (Lonza, Sigma) supplemented with 10% foetal bovine serum (FBS) (RPMI 1640 complete medium). Confluent THP-1 cells were harvested, seeded at $2 \times 10^5$ cells/per well in a 24 well plate. Cells were differentiated with phorbol 12-myristate 13-acetate (PMA, Sigma) at a final concentration of 50 ng/μl for 72 hours (hrs). Spent medium was aspirated and cells washed three times with pre-warmed PBS. RPMI 1640 complete medium was added to each well and plates incubated overnight at 37˚C.

 Starter cultures of each strain was prepared in 5 ml 7H9 OADC (Difco, BD) and sub-cultured to an $OD_{600nm}$ of 0.05 in 30 ml. Following incubation at 37˚C for 5 days, 10 ml bacterial

culture for each bacterial strain was centrifuged (Eppendorf 5810), at 4000 x g for 10 min. Bacterial cultures were washed 3 times in an equal volume of PBS and pellets resuspended in 5 ml RPMI 1640 complete medium. Resuspended cells were sonicated in a water bath sonicator for 12 min at 37˚C to disrupt cell clumps. Bacterial cultures were filtered through a 40 μm filter (Falcon) and $OD_{600nm}$ was measured. An established conversion factor of $OD_{600nm}$ 1.0 = 1 x $10^8$ bacteria was used to dilute bacterial cultures to obtain a multiplicity of infection (MOI) of 2:1 and 5:1.

THP-1 cells were infected for 3 hrs and treated with 1 ml 1:100 Pen/Strep (10 000 U/ml, Sigma) for 40–60 min to kill any extracellular bacteria. Pen/Strep was removed and the cells washed 3 times with PBS and replaced with RPMI 1640 complete medium. At the various time points 500 μl $dH_2O$ was added to the wells (triplicate) to lyse the THP-1 cells and release intracellular bacteria. Bacteria were enumerated by plating appropriate dilutions on solid media.

## Flow cytometry

Median fluorescent intensity (MFI) was measured by flow cytometry at early log (day 4), mid log (day 10) and early stationary phase (day 14) during *in vitro* growth and 0, 24 hrs, 48 hrs and 72 hrs during infection studies. The *Mycobacterium smegmatis* mc$^2$155 strain containing pCHARGE3 plasmid was used as positive control (S1 Table in S1 File). Culture aliquots or THP-1 lysates were sonicated in a water bath sonicator for 12 min at 37˚C to disrupt clumps and centrifuged (Eppendorf 5810) at 12 500 x g for 5 min. Pellets were resuspended in 200 μl 4% formaldehyde and fixed for 30 min in the dark. The fixed cultures were washed with 800 μl PBS containing 0.05% Tween-80, resuspended in 200 μl PBS-0.05% Tween-80 and stored at 4˚C until further processing. Prior to flow cytometry analysis, samples were centrifuged for 5 min at 12 500 x g, resuspended in 500 μl PBS and filtered through a 40 μm filter (Falcon). Analysis was performed using the FACSJazz (BD) where MFI was measured at an excitation of 510 nm and 610/20 filter and 30 000 events were recorded [22].

## Clinical assessment of SufR Rv1460 immunogenicity

**Ethics approval.** This study was approved by the Human Research Ethics Committee (HREC) of the Faculty of Medicine and Health Science, University of Stellenbosch, S17/02/45. Participant recruitment for the study is covered under studies: N16/05/070 and N14/10/136. We obtained informed written consent from each participant. No minors were included. The study in which protein were purified for antibody generation was covered by ethics S13/09/159.

**Study participant recruitment and sample collection.** Study participants were recruited in three Cape Town suburbs. Newly diagnosed untreated TB (n = 20), latently infected (IGRA positive) (n = 9) and healthy uninfected individuals (IGRA negative) (n = 17) were recruited for this study (S4 Table in S1 File). Since the proportion of healthy uninfected individuals recruited from the abovementioned communities was small this group of individuals were recruited from the Division of Molecular Biology and Human Genetics at the University of Stellenbosch.

Blood samples were collected in 9 ml sodium heparin (NaHep) tubes (Lasec) and processed within 2 hrs of collection. NaHep tubes (Lasec) were inverted 10X and 2.5 ml blood was removed for whole blood assay (WBA, 12-hour(hr) assay) and lymphocyte proliferation assay (LPA, 7-day assay).

**Recombinant SufR production.** Recombinant SufR was produced and purified as detailed in Willemse et. al. (2018). Briefly, 6xHis-tagged SufR was expressed in the *E. coli* Arctic express DE3 strain and purified by two rounds of Ni-IMAC affinity chromatography. Prior

to the second HiTrap column, the recombinant protein was concentrated by ultra-filtration, dialysed to remove imidazole and the 6xHis-tag removed by tabaco etch virus protease cleavage. The protein was again concentrated by ultrafiltration and further purified by gel filtration chromatography.

**Phenotype screening using whole blood assay and lymphocyte proliferation assay.** Whole blood assay was done as described by Loxton *et al* (2017). Briefly 500 μl heparinized whole blood were added to tubes that contained 5 μl anti-CD28 and anti-CD49 (0.5 μg/ml) (BD Biosciences) co-stimulatory antibodies. To each of the tubes, 100 μl of following stimulants were added: RPMI/L-Glutamine; *M. tuberculosis* recombinant protein SufR final concentration of 1 μg/ml, BCG (SSI, Denmark) final concentration of $1 \times 10^6$ CFU/ml and Phytohaemagglutinin (PHA, Bioweb) final concentration of 5 μg/ml, respectively. The tubes were vortexed and incubated at 37˚C in a humidified incubator (ESCO Vivid Air) with 5% $CO_2$ for 12 hrs. Two supernatant aliquots (55 μl) were harvested and stored at -80˚C for Luminex Multiplex analysis whereafter Brefeldin A (Sigma, 5 μg/ml) was added, vortexed and incubated in a 37˚C water bath for 5 hrs. Lymphocytes were harvested and stored at -80˚C for phenotype screening [23].

Lymphocyte proliferation assay was done as described by Loxton *et al* (2017). Briefly heparinized whole blood was diluted 1:10 in RPMI 1640 supplemented with 1% L-Glutamine and 1.150 ml of the diluted whole blood was incubated in a 24-well plate with 100 μl each of RPMI/L-Glutamine, *M. tuberculosis* recombinant protein SufR final concentration of 1 μg/ml, BCG (SSI, Denmark) final concentration of $1.25 \times 10^6$ CFU/ml and PHA (Bioweb) final concentration of 1 μg/ml (added on day 4), respectively. Two supernatant aliquots (250 μl) were harvested and stored at -80˚C for Luminex Multiplex analysis. Cells were further incubated with Brefeldin A (Sigma, final concentration of 2 μg/ml), PMA (Sigma, final concentration of 20 ng/ml) and ionomycin (Sigma, final concentration of 2 μg/ml) for 4 hrs at 37˚C. PMA and ionomycin was added to all the well except the unstimulated control. Lymphocytes were harvested and stored at -80˚C for phenotype screening [23].

**Luminex multiplex immuno assay.** The levels of 8 selected cytokines (Interferon gamma (IFN-g), Monocyte chemoattractant protein-1 (MCP-1) (CCL2), Macrophage inflammatory protein (MIP)-1b, Interleukin (IL)-1b, Matrix metallopeptidase (MMP)-9, Interleukin (IL)-6, Interleukin (IL)-8 and regulated and normal T cell expressed and secreted (RANTES) (CCL5)) were measured in the WBA and LPA supernatants using an 8-plex analyte-specific antibody kit (Luminex Human Magnetic Assay, R&D Systems). All samples were diluted 1:2 and the assay were done according to the Manufacturer's instructions. The assays were performed and read in a central laboratory on the Bio-Plex platform (Bio-Plex® MAGPIX™ Multiplex reader (MAGPIX13046704, BioRad Laboratories, Inc., California, USA).), with the Bio-Plex Manager Software V.6.1 used for bead acquisition and analysis.

**Flow cytometry.** Cryopreserved lymphocytes from the WBA and LPA were washed in 10 ml PBS and centrifuged (Eppendorf 5810) for 10 min at 400 x g. The pellet was resuspended in 200 μl 1X permeabilization buffer (Biolegend), incubated at room temperature for 10 min and centrifuged (Eppendorf 5810) for 10 min at 400 x g. The cells were stained with 20 μl antibody cocktail: cluster differentiation (CD3); CD4 CD8; CD19; IFN-g; tumour necrosis factor alpha (TNF-a); IL-2; IL-10 (S3 Table in S1 File) and incubated at 4˚C (in the dark) for 1 hr. Cells were washed 2X with 1X permeabilization buffer and resuspended in 200 μl PBS and acquired on the BD FACS CANTO II Flow cytometer (BD Biosciences). A total of $10^5$ to $10^6$ cells were analysed per tube.

**Statistical and data analysis.** GraphPad Prism 8 software was used for the statistical analysis of the molecular biology protein concentrations and expression analysis. Multiple t-test was done without assuming equal standard deviation among samples and without correction

for multiple comparisons. A *p*-value of smaller than 0.05 was considered statistically significant. Analysis of the immunogenicity of SufR was performed by a qualified statistician (Prof Martin Kidd) using the Statistica 13 software. For all analysis and comparisons, a *p*-value of smaller than 0.05 was considered statistically significant.

## Results

### Development of *sufR* reporter

Previous studies have demonstrated that the *suf* operon is induced by oxidative and nitrosative stress, iron limitation and intracellular growth [24, 25]. To further explore *suf* operon regulation, we sought to develop a fluorescent reporter that could be used to monitor *sufR* expression at a single cell level. To this end, we created a reporter construct by fusing the promoter region of *sufR* to the fluorescent protein mCherry and cloning it into an integrating plasmid backbone (pMV306_123mCherry). Growth of *M. tuberculosis* transformed with the reporter construct *in vitro* was indistinguishable from the wild wt control strain transformed with the plasmid backbone (pMV306) (Fig 1A), confirming *mCherry* expression did not alter growth. Flow cytometry analysis of the fluorescence per cell showed a significant increase in the MFI in the fluorescent reporter strain when compared to the negative control (wt, pMV306 empty vector) (Fig 1B) at days 4, 10 and 14, while no difference in the MFI between the respective days was observed.

In order to validate the use of fluorescence reporter strains as tool for the quantification of gene expression, we determined *mCherry* and *sufR* expression at days 4, 10 and 14 (Table 1). The *sufR* transcript levels for both the H37Rv_*attB*::pMV306 and H37Rv_*attB*::pMV306_123mCherry strain increased between day 4 and day 10, while a decrease in transcript levels was observed between day 10 and 14. The *mCherry* transcript levels for H37Rv_*attB*::pMV306_123mCherry strain decreased between day 4 and day 10 (1.8fold), and between day 10 and 14 (3.4fold). As expected, no *mCherry* expression was observed in the control strain. The ratio of *mCherry* transcript to *sufR* transcript was not consistent at the 3 time points, suggesting that *mCherry* transcript levels do not reflect the *sufR* transcript levels.

### Monitoring *sufR* induction during intracellular growth

Although we did not see a correlation between *mCherry* and *sufR* transcript levels, we reasoned that the reporter could still be useful for monitoring promoter activity during intracellular

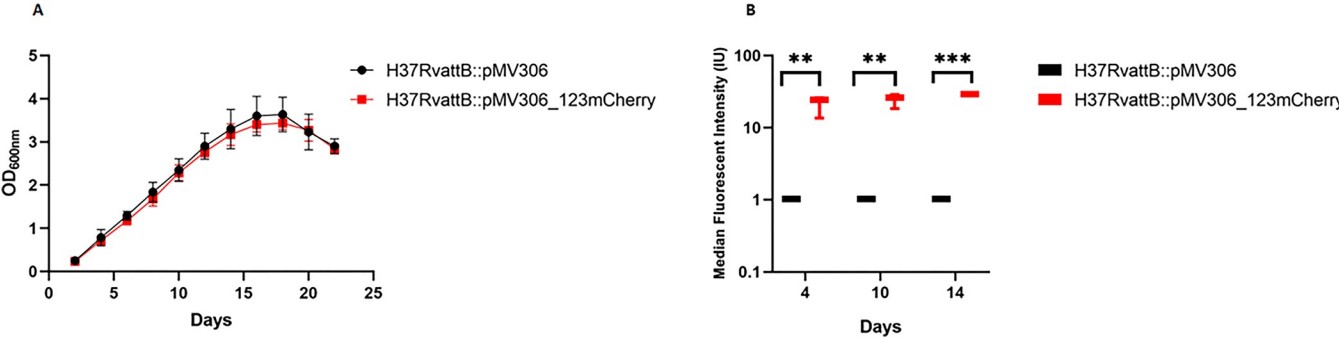

**Fig 1. Growth and fluorescent intensity monitored under standard culture conditions.** A) Growth curves of *M. tuberculosis* H37Rv_*attB*::pMV306 (wt control) and *M. tuberculosis* H37Rv_*attB*::pMV306_123mCherry (fluorescent reporter) strains measuring the $OD_{600nm}$ over the course of 22 days. Each point is the average and standard deviation of three biological replicates. B) MFI of *M. tuberculosis* H37Rv_*attB*::pMV306 (wt) and *M. tuberculosis* H37Rv_*attB*::pMV306_123mCherry (fluoresecent reporter) strains measuring the expression (MFI) of mCherry over time at day 4, day 10 and day 14. The relative fluorescence intensity of the two strains was measured in a BD FACSJazz at 510 nm using a 610/20 filter over time. The increase in fluorescence is a measure of the increase in *sufR* expression overtime. Each point is representative of three biological and technical replicates. The star indicates the significant difference and levels of significance between MFI of the different strains as determined by multiple unpaired *t*-test. $p \leq 0.05$ *, $p \leq 0.01$ **, $p \leq 0.001$ *.

**Table 1. Relative expression of *sufR* and *mCherry*.**

| | Mean ± SD | | |
|---|---|---|---|
| | **Day 4** | **Day 10** | **Day 14** |
| ***sufR/sigA*** | | | |
| H37Rv_*attB*::pMV306 | 77.96 ± 17.78 | 99.32 ± 51.23 | 12.42 ± 4.01 |
| H37Rv_*attB*::pMV306_123mCherry | 23.32 ± 24.38 | 82.90 ± 35.59 | 7.24 ± 2.28 |
| ***mCherry/sigA*** | | | |
| H37Rv_*attB*::pMV306 | 0.00 ± 0.00 | 0.00 ± 0.00 | 0.00 ± 0.00 |
| H37Rv_*attB*::pMV306_123mCherry | 0.56 ± 0.19 | 0.31 ± 0.08 | 0.09 ± 0.06 |
| ***sufR/mCherry*** | | | |
| H37Rv_*attB*::pMV306_123mCherry | 56.06 ± 40.66 | 319.73 ± 227.83 | 102.21 ± 38.18 |

growth. No significant difference in the uptake of *M. tuberculosis* H37Rv_*attB*::pMV306 and H37Rv_*attB*::pMV306_123mCherry by the THP-1 cells was observed for both MOI investigated (2:1 and 5:1) (Fig 2A). No significant difference in the survival of the two strains was observed, while a lower bacterial load was observed for the higher MOI (Fig 2B and 2C), which might be due to increased death of the THP-1 cells because of the high bacterial burden.

As expected, we saw no fluorescence for the negative control at the 4 time points for both MOIs. For the MOI of 2:1 (Fig 3A) at 0 hrs, the histogram for H37Rv_*attB*::pMV306_123mCherry strain coincided with the wt control (H37Rv_*attB*::pMV306). At 24 hrs, a small shoulder appears to the right of the negative population for the fluorescent strain,

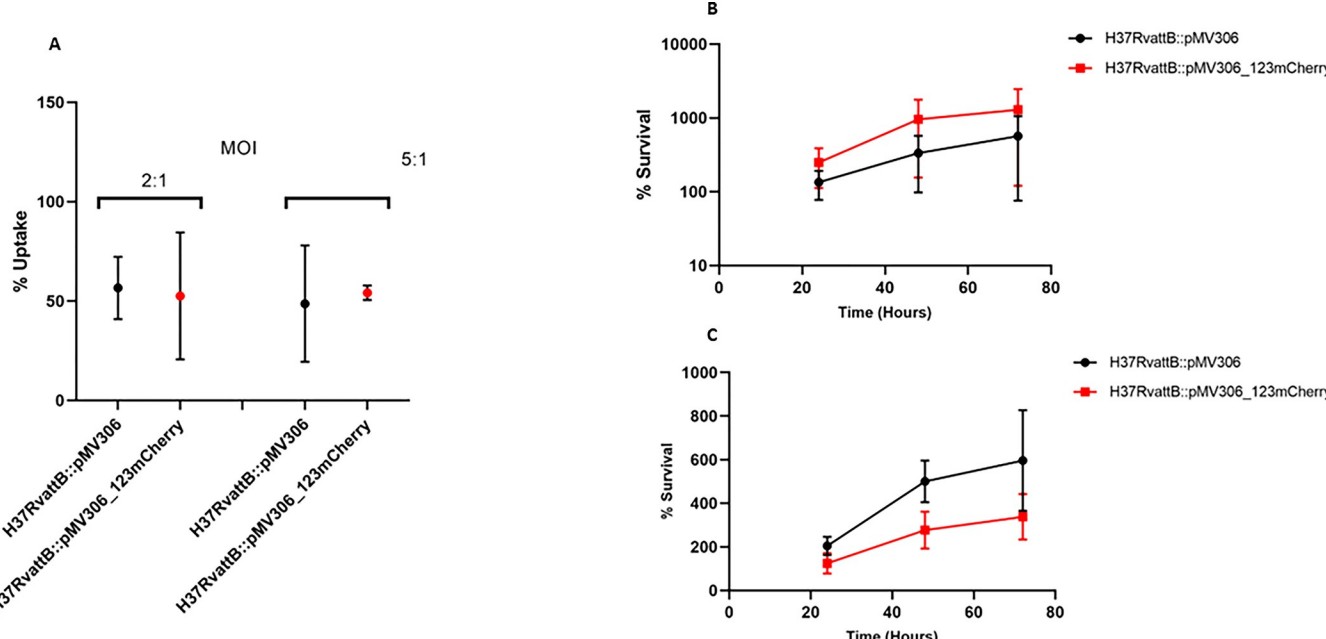

**Fig 2. *M. tuberculosis* H37Rv control and fluorescent reporter strain uptake and survival within differentiated THP-1 macrophages.** THP-1 macrophages were infected with *M. tuberculosis* H37Rv_*attB*::pMV306 (wt control) and *M. tuberculosis* H37Rv_*attB*::pMV306_123mCherry (fluorescent reporter) at a MOI of 2:1 and 5:1 for 3 hrs. A) % Uptake is represented by the number of bacterial cells internalized determined by dividing CFU/ml count of the original cultures (MOI: 2:1 and 5:1) by the CFU/mL count of the bacterial culture at time 0 (3hrs after infection).Pertentage survival within differentiated THP-1 macrophages at a MOI of B) 2:1 and C) 5:1. THP-1 cells were lysed, and bacterial cells were harvested, and serial dilutions plated on solid media at 24, 48 and 72 hours post infection, to determine CFU/ml. Each point is representative of three biological and three technical replicates. No significant difference in the survival of the bacteria of the two strains as determined by multiple unpaired *t*-test.

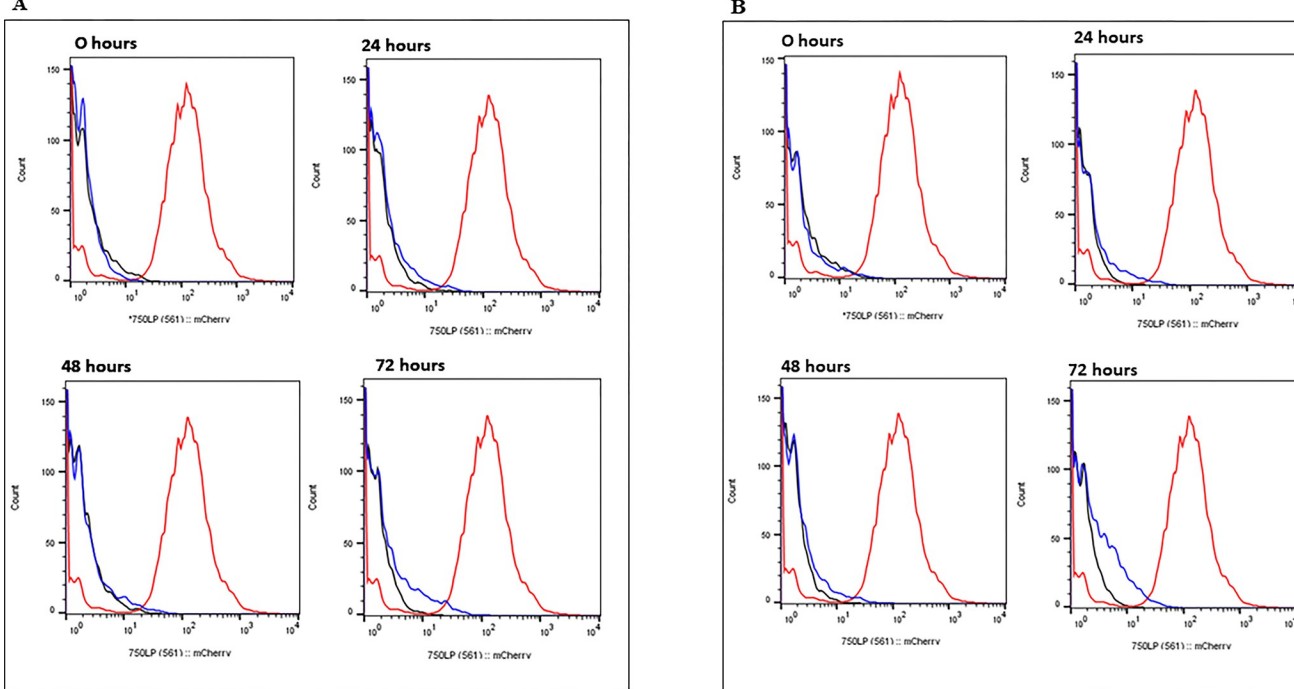

**Fig 3. *sufR* promoter activity during intracellular growth in THP-1 cells.** *M. tuberculosis* H37Rv THP-1 infection with a MOI of A) 2:1 and B) 5:1. *M. tuberculosis* H37Rv_*attB*::pMV306 (wt control) (black) and *M. tuberculosis* H37Rv_*vattB*::pMV306_123mCherry (fluorescent reporter)(blue) strains was used to infect THP-1 macrophage like cells, lysed at 0, 24, 48, and 72 hrs post-infection. *M. smegmatis* strain containing a plasmid expressing mCherry was used as positive control (red). The bacterial culture was harvested and fixed with 4% formaldehyde and run-on BD FACSjazz flow cytometer to measure the mCherry fluorescence (510 nm with 610/20 filter) which is a measure of *sufR* gene expression.

and this shoulder continued to grow at 48 and 72 hrs. This suggests that a subset of the population of bacteria within the macrophages has higher expression of the reporter gene. For the MOI of 5:1 (Fig 3B), the H37Rv_*attB*::pMV306_123mCherry strain shows a shoulder to the right of the negative population at 24 and 48 hrs. At 72 hrs a significant shift to right is seen for H37Rv_*attB*::pMV306_123mCherry strain, when compared to the negative control. This shift at 72 hrs by the fluorescent strains at a MOI of 5:1 is the greatest shift seen for all time points and both MOI's. Taken together, these results suggest that *sufR* expression is induced in a subset of bacteria within macrophages, and this population increases over the course of a 72-hr infection.

## Immunogenicity of SufR protein

Two different assays were performed to assess the immunogenicity of the transcription regulator SufR: 1) a short term 12-hr stimulation to characterise the production of IFN-g and other cytokines/growth factors suggestive of an effector response and 2) a longer term 7-day stimulation to see if SufR induces a memory type immune response. The levels of 8 whole blood supernatant analytes, MCP-1, RANTES, IL-1b, IL-8, MIP-1b, IFN-g, IL-6 and MMP-9 were measured in the clinical groups: active TB, QFN pos and QFN neg. The clinical and demographical characteristics of study participants is shown in S4 Table in S1 File.

## 12-hour stimulation with SufR and BCG

The gating strategy of the phenotype screening of the T and B cell are shown in S1 Fig in S1 File. Low levels of MCP-1, IL-1b, MIP-1b, IFN-g and IL-6 was observed in all three clinical

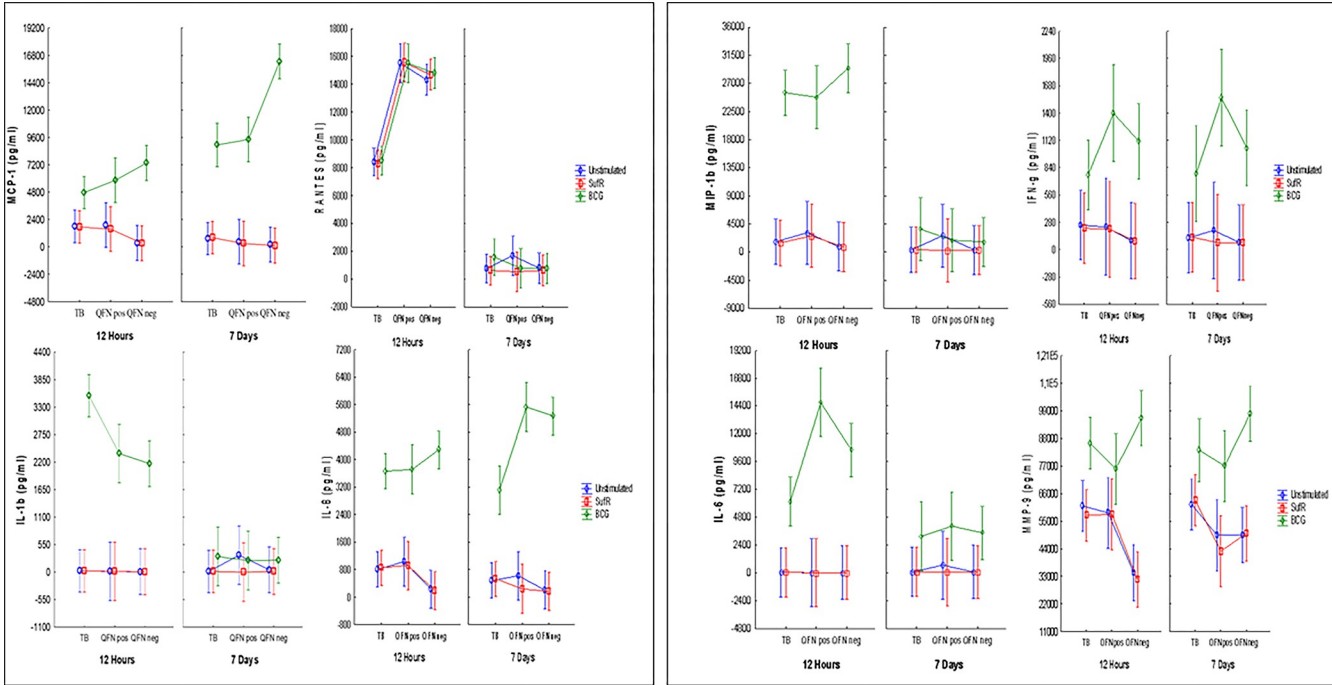

**Fig 4. Luminex analysis of analyte concentrations in whole blood.** Whole blood from participants (active TB (n = 20), QFN pos (n = 9) and QFN neg (n = 17) groups) was left unstimulated or stimulated with SufR and BCG for 12 hrs or 7 days. Supernatant analytes (pg/ml) of (A) MCP-1, (B) RANTES, (C) IL-1b, (D) IL-8, (E) MIP-1b, (F) IFN-g, (G) IL-6 and (H) MMP-9 were measured. ANOVA test was performed to determine statistically significant differences between groups. Vertical bars denote 0.95 CI.

groups following 12 hrs of stimulation by SufR (Fig 4). Lower levels of IL-8 although not significant is seen in the QFN neg group when compared to QFN pos ($p = 0.06$) and active TB ($p = 0.08$) groups in the unstimulated and SufR stimulation, respectively. Significantly lower levels of MMP-9 were seen in the QFN neg group when compared to QFN pos ($p = 0.00$) and active TB ($p = 0.01$) groups when left unstimulated or SufR stimulated, respectively. Significantly lower levels of RANTES were measured in the active TB group unstimulated ($p = 0.00$ and $p = 0.00$), SufR stimulated ($p = 0.00$ and $p = 0.00$) or BCG stimulated ($p = 0.00$ and $p = 0.00$) when compared to QFN pos and QFN neg groups, respectively. BCG stimulation gave significantly higher levels for all analytes except RANTES in all the groups when compared to the unstimulated and SufR stimulated conditions. QFN neg group had the highest levels of MCP-1 compared to active TB ($p = 0.02$) and MMP-9 when compared to QFN pos ($p = 0.03$) following BCG stimulation. Active TB group had the highest levels of IL-1b when compared to QFN pos ($p = 0.00$) and QFN neg ($p = 0.00$) following BCG stimulation. Interestingly, IFN-g levels were the highest in the QFN pos group compared to active TB ($p = 0.00$) and IL-6 when compared to active TB ($p = 0.00$) and QFN neg ($p = 0.03$) following BCG stimulation. *Ex vivo*, 12-hr stimulation of whole blood from active TB, QFN pos and QFN neg groups with SufR elicits a measurable immune response for the selected analytes, even though not significant.

## 7-day restimulation with SufR and BCG

SufR induced very low levels of all measured cytokines in the active TB, QFN pos and QFN neg groups (Fig 4). BCG stimulation induced significantly higher levels of MCP-1, IL-8, IFN-g and MMP-9 in all groups when compared to the unstimulated and SufR stimulated populations. Additionally, BCG stimulation resulted in significantly higher levels in the QFN pos

group when compared to unstimulated (*p* = 0.04) and SufR stimulated (*p* = 0.04). Similar to the 12-hr BCG stimulation, the QFN neg group had the highest level of MCP-1 compared to active TB (*p* = 0.00) and QFN pos (*p* = 0.00) groups and MMP-9 when compared to the QFN pos (*p* = 0.00). The level of IL-8 and IFN-g was significantly higher in the QFN pos group compared to active TB group (*p* = 0.00 and *p* = 0.00) for BCG stimulation. IFN-g and MMP-9 levels showed similar patterns following 12-hr vs 7-day BCG stimulations (Fig 4).

## Phenotypic analysis of whole blood following SufR restimulation

We evaluated the phenotype of CD4+, CD8+ and CD19+ cells following stimulation of whole blood from the active TB, QFN pos and QFN neg groups. The frequency of four cytokines IFN-g, TNF-a, IL-2, and IL-10 was measured in all the cell subtypes. Combinations of these four cytokines were also measured analysed to determine the frequency of multifunctional T and B cells under the different stimulation conditions at the 12-hr and 7-day time points.

## 12-hour or 7-day SufR stimulation does not induce significant cytokine secreting CD4 T cells

There was no difference in the frequency of CD4+IFN-g+, CD4+TNF-a+, CD4+IL-2+, CD4+IL-10+ populations (Fig 5) in the different groups after 12 hrs of stimulation. Significantly increased frequencies of CD4+IFN-g+ and CD4+IL-2+ was observed for the QFN neg (*p* < 0.01; *p* < 0.01) group when compared to active TB after 12-hr PHA stimulation. Significantly higher frequencies of CD4+TNF-a+ was measured in the QFN neg group (p = 0.05) compared to active TB after 12-hr BCG stimulation and QFN pos group (*p* < 0.01) and QFN neg group (*p* < 0.01) compared to active TB after 12-hr PHA stimulation. No differences were measured in the CD4+ single cytokine frequencies following 12-hr or 7-day SufR stimulation (Fig 5). Increased frequencies of CD4+TNF-a+ was measured in the active TB (*p* < 0.01), QFN pos (*p* < 0.01) and QFN neg groups (*p* < 0.01) after 7-day stimulation with SufR and BCG when compared to the unstimulated populations (Fig 5). Table 2 lists the frequencies of the CD4+ multifunctional subsets after 12-hr and 7-day stimulation with BCG and PHA.

## CD8+ T cell phenotypes are poorly stimulated by SufR but cytokine-secreting B cells are increased

CD8+ T cell phenotypes were not different between groups when stimulated with SufR in the short term or 7-day assays (S2 Fig and S5 Table in S1 File). Similarly, the CD19+ T cell phenotypes were not different between groups when stimulated with SufR in the short term or 7-day assays (S3 Fig in S1 File), while multifunctional CD19+IFNg-IL2+IL10-TNFa- B cells were significantly increased in the QFN pos group (*p* = 0.02) when compared to active TB following 12-hr SufR stimulation (S6 Table in S1 File). We also observed significantly higher levels of CD19+IFNg-IL2-IL10-TNFa+ cells in the QFN neg group (*p* = 0.02) when compared to active TB following 7-days SufR stimulation (S6 Table in S1 File).

## Discussion

*M. tuberculosis*, a human adapted pathogen has found multiple ways to manipulate the host immune response during and after infection [26]. The human immune response to *M. tuberculosis* infection is a highly complex cascade of reactions [27], with macrophages as the preferred intracellular location [28]. Within the alveolar macrophages several mechanisms are employed to kill the bacilli, such as vacuole acidification, proteases, antimicrobial peptides, reactive oxygen, nitrogen species and changes in ion flux [29]. Interaction with the host

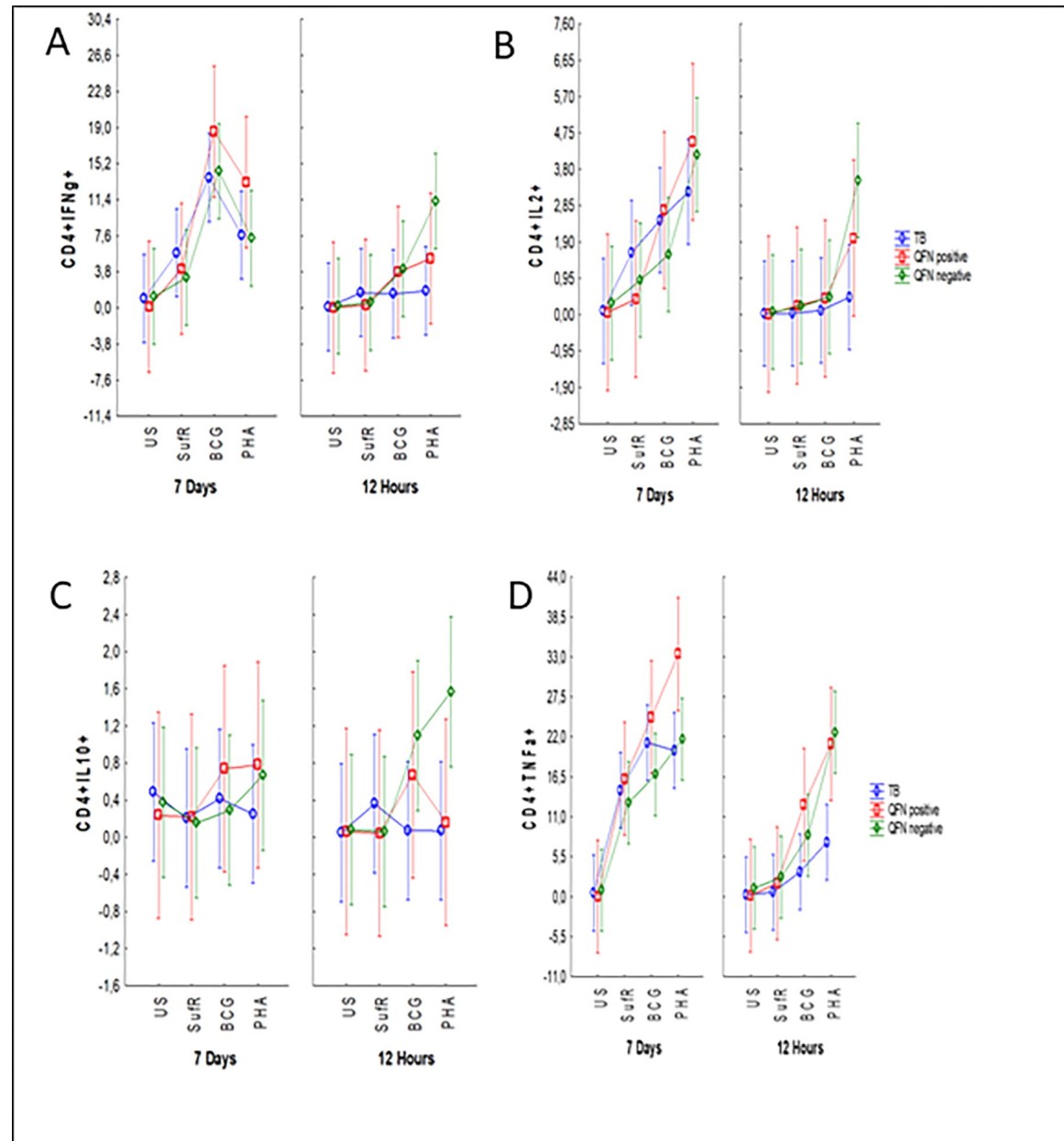

**Fig 5. Flow cytometry phenotype screening of CD4+ T cell subsets in whole blood.** The graph shows the CD4+ T cell subsets in whole blood from active TB group (n = 20), QFN pos group (n = 9) and QFN neg group (n = 17) after 12 hours and 7 days of being unstimulated and stimulated with SufR, BCG and PHA. (A) CD4+IFN-g, (B) CD4+IL-2+, (C) CD4+IL-10+ and (D) CD4+TNF-a+. ANOVA test was performed to determine statistically significant differences between groups. Vertical bars denote 0.95 CI.

through infection gives rise to expression of specific gene products for survival and multiplication within the host. The *M. tuberculosis* SUF system has been implicated in the pathogen's intracellular survival, due to its role in Fe-S cluster biosynthesis and repair [16].

In this study a fluorescence reporter was used to monitor *sufR* expression at a single cell level during intracellular growth. In agreement with Carroll *et al* (2010), recombinant mCherry expression in *M. tuberculosis* did not alter fitness *in vitro* and in macrophages, supporting its utility as a reporter in mycobacteria [30]. Comparison of *sufR* and *mCherry*

**Table 2. List of multifunctional CD4+T cell subset frequencies of in active TB, QuantiFERON negative and QuantiFERON positive groups following BCG and PHA stimulation at 2 timepoints.**

| | | 12-hour | | | 7-day | | |
|---|---|---|---|---|---|---|---|
| | | Active TB VS QFN pos | Active TB VS QFN neg | QFN neg VS QFN pos | Active TB VS QFN pos | Active TB VS QFN neg | QFN neg VS QFN pos |
| CD4+IFNg+IL2+IL10 +TNFa+ | BCG | Ns | Ns | ns | $p = 0.01$ | ns | $P = 0.00$ |
| | PHA | Ns | $p = 0.01$ | $p = 0.05$ | ns | ns | ns |
| CD4+IFNg+IL2 +IL10-TNFa+ | BCG | Ns | Ns | ns | ns | ns | ns |
| | PHA | Ns | Ns | ns | ns | ns | ns |
| CD4+IFNg+IL2 +IL10-TNFa- | BCG | Ns | Ns | ns | ns | $p = 0.00$ | $p = 0.02$ |
| | PHA | Ns | Ns | ns | ns | ns | ns |
| CD4+IFNg+IL2-IL10 +TNFa+ | BCG | Ns | Ns | ns | ns | ns | $p = 0.05$ |
| | PHA | Ns | $p = 0.00$ | $p = 0.02$ | ns | ns | ns |
| CD4+IFNg +IL2-IL10-TNFa+ | BCG | Ns | Ns | ns | ns | ns | $P = 0.03$ |
| | PHA | Ns | Ns | ns | $p = 0.05$ | ns | $p = 0.04$ |
| CD4+IFNg +IL2-IL10-TNFa- | BCG | Ns | Ns | ns | ns | $p = 0.01$ | $p = 0.02$ |
| | PHA | Ns | $p = 0.01$ | ns | ns | ns | ns |
| CD4+IFNg-IL2 +IL10-TNFa+ | BCG | Ns | Ns | ns | ns | ns | ns |
| | PHA | $p = 0.04$ | $p = 0.00$ | ns | $p = 0.03$ | $p = 0.01$ | ns |
| CD4+IFNg-IL2 +IL10-TNFa- | BCG | Ns | Ns | ns | ns | ns | ns |
| | PHA | Ns | $p = 0.00$ | ns | ns | $p = 0.00$ | $P = 0.00$ |
| CD4+IFNg-IL2-IL10 +TNFa+ | BCG | Ns | Ns | ns | ns | ns | ns |
| | PHA | Ns | $p = 0.01$ | ns | $p = 0.02$ | ns | ns |
| CD4+IFNg-IL2-IL10 +TNFa- | BCG | Ns | $p = 0.04$ | ns | ns | ns | ns |
| | PHA | Ns | Ns | ns | ns | ns | ns |
| CD4+IFNg-IL2-IL10-TNFa + | BCG | $p = 0.00$ | Ns | ns | ns | ns | ns |
| | PHA | $p = 0.00$ | $p = 0.00$ | ns | ns | $p = 0.01$ | $p = 0.00$ |
| CD4+IFNg- IL2-IL10-TNFa- | BCG | Ns | Ns | ns | ns | ns | ns |
| | PHA | Ns | $p = 0.00$ | ns | $p = 0.05$ | ns | ns |

transcript levels during *in vitro* growth revealed that *mCherry* transcript levels were much lower than *sufR* transcript levels, and the ratio of *sufR/mCherry* transcript did not remain constant over time in the fluorescent strain. A possible explanation for this is differences in the transcript stability for *mCherry* and *sufR*. A study analysing transcript stability in *M. tuberculosis* showed that the mean transcript stability was 9.5 min, which is significantly longer than most other prokaryotes [31]. A lower transcript stability for *mCherry* mRNA could account for the lower transcript levels observed at all time points and result in an increase in the ratio of *sufR/mCherry* when *sufR* transcript levels rise in mid-log phase. In addition, a decrease in the *mCherry* transcript levels was observed between day 4 and 14, while the MFI remained constant for the reporter strain at all time points. Carroll *et al* (2010) demonstrated that mCherry is very stable with fluorescence maintained over 12 days after protein synthesis was inhibited by chloramphenicol [30]. Therefore, decreased *mCherry* expression at days 10 and 14 appears to be masked by the long half-life of the mCherry protein. As such, the reporter is useful for measuring cumulative promoter activity rather than expression at a specific time point.

To evaluate *sufR* promoter activity during intracellular growth, we infected a THP-1 macrophage-like cells with the H37Rv fluorescent reporter strain and an empty vector control. Our CFU results show no statistically significant differences in number of bacteria that were phagocytosed for either of the MOI's (2:1 and 5:1). Lower survival of the bacteria within the host cells was observed for the higher MOI, which might be due to increased death of the THP-1

cells because of the high bacterial burden (Fig 2C). This is consistent with a study by Bettencourt *et al* (2017) showing that increasing MOI leads to a decrease in THP-1 cell viability [32].

In the first 24 hrs post infection, bacterial numbers are relatively stable, while an increase in CFU between 24 and 48 hrs was observed, indicative of replication. Raffetseder *et al* (2014) found that infection of human monocyte-derived macrophages (hMDMs) with *M. tuberculosis* H37Rv at a lower MOI of 1 showed no significant increase in replication for at least 10 days, a period during which the viability of infected and uninfected cells was similar. The higher MOI of 10 showed significant increase in bacterial growth by day 7, followed by extensive cell death and increasing numbers *of M. tuberculosis* in the extracellular fraction [33].

*In vivo* flow cytometric analysis of intracellular bacteria (Fig 3A and 3B) revealed increased fluorescence in a small subset of the bacteria harbouring the reporter relative to the control strain at 24, 48 and 72 hours. The greatest increase in fluorescence was observed at the 72-hr time point for both MOIs, with the greatest increase seen for the higher MOI. Our results suggest that *sufR* expression is induced in a subset of bacteria within macrophages, and this population increases over the course of a 72-hr infection. Previous studies have demonstrated population heterogeneity for *M. tuberculosis* growing intracellularly, where sub-populations displaying reduced replication were detected [22, 34]. Techniques that allow gene expression to be monitored at a single cell level are therefore essential when studying intracellular *M. tuberculosis*, as measuring changes in average expression may be unable to detect differences in these minor sub-populations.

We hypothesized that SufR is expressed during intracellular growth and human infection and induces an immune response that confers diagnostic potential. Therefore, we assessed the immunogenicity of SufR by direct *ex vivo* stimulation of whole blood samples from newly diagnosed and untreated active TB and QFN pos and QFN neg participants. We assessed the immunogenicity of SufR by measuring the levels of 8 whole blood supernatant analytes MCP-1, RANTES, IL-1b, IL-8, MIP-1b, IFN-g, IL-6, and MMP-9. Higher levels of IL-8 ($p = 0.08$) although not significant and significantly higher levels of MMP-9 ($p = 0.00$) in the active TB group when compared to the QFN neg group for the short-term stimulation (12 hrs) with SufR is suggestive of an effector or memory response. However, this response was not unique to SufR stimulation, similar results was shown for the unstimulated population. Krupa *et al* 2015 has shown significantly higher IL-8 levels in BAL Fluids ($p < 0.001$) and plasma ($p < 0.02$) from TB patients than in normal subjects [35]. MMP's are not expressed under normal circumstances but their overexpression is observed during inflammation. An increase in MMP-8 and MMP-9 are seen in pulmonary TB and reflects the severity of the destructive process [36]. RANTES is involved in the selective attraction of memory T cells [37], a signalling molecule that is produced by T cells, macrophages, and a variety of cell types. Lee *et al* (2008) suggest that altered RANTES production might play a valuable role in immunopathogenesis during TB [38]. We showed significantly lower levels of RANTES in the active TB group after 12 hrs when left unstimulated ($p = 0.00$ and $p = 0.00$), SufR stimulated ($p = 0.00$ and $p = 0.00$) or BCG stimulated ($p = 0.00$ and $p = 0.00$) when compared to QFN pos and QFN neg groups, respectively. A study done by Teklu *et al* (2018) showed quantitative changes for RANTES, both antigen stimulated and unstimulated blood plasma supernatants, in active TB compared to LTBI cases, [37]. Our results for stimulated and unstimulated is similar, except that the levels of RANTES in our active TB group is significantly decreased when compared with LTBI and uninfected control group.

We evaluated conventional T cell subsets, and CD19+ B cell responses using flow cytometry analysis to see if these phenotypes can distinguish between active TB, LTBI and healthy controls. A study done by Lichtner *et al* (2015) showed slightly higher frequency of both "all IFN-g +" and "all IL-2+" CD4+ T cells in LTBI patients with respect to active TB patients [39].

Stimulation with SufR did not alter the CD4+IFN-g+, CD4+IL-2+, CD4+TNF-a+ and CD4 +IL-10+ frequencies in a way that would enable us to distinguish between groups. Although not significant, long-term stimulation with of SufR showed higher frequencies of CD4+IFNg +IL2+IL10-TNFa- multifunctional T cells in active TB group when compared to QFN neg ($p = 0.15$) and QFN pos ($p = 0.021$) group, while short term stimulation showed higher frequencies in CD4+IFNg-IL2+IL10-TNFa- subset in QFN neg group when compared to QFN pos ($p = 0.05$) group. Similar to our finding, Caccamo *et al* (2010) showed (short term stimulation) the percentage of CD4+ cells producing both IFN-g+ and IL-2+ was significantly increased in TB patients compared to LTBI subjects [40]. Analysis of B cells shows higher frequency of CD19+IL-2+ (not significant) in the QFN pos group ($p = 0.07$) when compared to active TB following 12-hr SufR stimulation. There were also significantly higher levels of CD19 +TNF-a+ in QFN neg group ($p < 0.01$) when compared to active TB following 12-hr PHA stimulation.

The cytokine response elicited by SufR for both the short and longer term restimulation assays were low and did not seem to induce a strong discriminatory potential according to the phenotypic screening. Immune recognition play an important role during the production of an immune response. Recombinant protein production in *E. coli* may result in improper protein folding or the lack of post-translational modifications, thereby reducing recognition and the resulting immune response to SufR. In the case for the active TB participants, several studies have shown that even though participants have active disease, they have lower immune responses to Mtb antigens for a variety of reasons, including migration of antigen to site of disease and increased cell death during active disease [41, 42]. In addition, the cytokines selected may not be part of the response pathway to SufR, and therefore future studies should include broad and unbiased cytokine response analysis.

## Conclusion

Infection studies revealed *sufR* promoter induction in a subset of intracellular *M. tuberculosis* between 24 and 72 hours. The immune response elicited by SufR for both whole blood assay and lymphocyte proliferation assay were low and phenotypic screening suggests that it does not have the potential to discriminate between the groups investigated.

## Supporting information

**S1 File.**
(DOCX)

## Acknowledgments

The authors acknowledge Dr Danicke Willemse for expressing and purifying the SufR protein used in this study. Dr Nasiema Allie for helping with Biosafety level-3 training and competency. All the study participants and clinical staff for collecting clinical samples. Professor Martin Kidd for statistical analysis. Dr Jomien Mouton, for providing the *Mycobacterium smegmatis* pCherry plasmid.

## Author Contributions

**Conceptualization:** Lucinda Baatjies, Andre G. Loxton, Monique J. Williams.

**Formal analysis:** Lucinda Baatjies, Andrea Gutschmidt, Andre G. Loxton, Monique J. Williams.

**Funding acquisition:** Andre G. Loxton, Monique J. Williams.

**Investigation:** Lucinda Baatjies, Ilana C. van Rensberg, Candice Snyders, Andrea Gutschmidt, Andre G. Loxton, Monique J. Williams.

**Methodology:** Andre G. Loxton, Monique J. Williams.

**Project administration:** Monique J. Williams.

**Resources:** Andre G. Loxton.

**Writing – original draft:** Lucinda Baatjies.

**Writing – review & editing:** Andre G. Loxton, Monique J. Williams.

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
