## [Decision Letter · Decision Letter 0]

21 Mar 2023

PONE-D-23-04734Investigating Mycobacterium tuberculosis sufR (rv1460) in vitro and ex vivo expression and immunogenicityPLOS ONE

Dear Dr. Williams,

Thank you for submitting your manuscript to PLOS ONE. After careful consideration, we feel that it has merit but does not fully meet PLOS ONE’s publication criteria as it currently stands. Therefore, we invite you to submit a revised version of the manuscript that addresses the points raised during the review process.

We look forward to receiving your revised manuscript.

Kind regards,

Wenping Gong, Ph.D.

Academic Editor

PLOS ONE

Journal Requirements:

Reviewers' comments:

Reviewer's Responses to Questions

**Comments to the Author**

1. Is the manuscript technically sound, and do the data support the conclusions?

Reviewer #1: Yes

Reviewer #2: Yes

Reviewer #3: Partly

Reviewer #4: Yes

2. Has the statistical analysis been performed appropriately and rigorously? 

Reviewer #1: Yes

Reviewer #2: Yes

Reviewer #3: N/A

Reviewer #4: Yes

3. Have the authors made all data underlying the findings in their manuscript fully available?

Reviewer #1: Yes

Reviewer #2: Yes

Reviewer #3: Yes

Reviewer #4: Yes

4. Is the manuscript presented in an intelligible fashion and written in standard English?

Reviewer #1: Yes

Reviewer #2: Yes

Reviewer #3: Yes

Reviewer #4: Yes

5. Review Comments to the Author

Reviewer #1: It is good to dig into other laboratory tests to accelerate the way towards fast and cheaper diagnostic techniques. It is good to know that negative findings are important as positive findings. Anyhow, the study conduct matched scientific research methods.

Reviewer #2: 1.Kindly follow the similar version while using gene names eg., mcherry and sufR throughout the manuscript

2. The conclusion is missing in this study and highlights the novelty in it.

3. Kindly explain the demographic profiles of study participants, sample size and calculation.

4. Highlight the concentration of cells (eg., per ml) used for FACS study

5. Try to highlight the references in discussion section and not in the results (line 211 to 212)

Reviewer #3: The present study deals with the analysis of SufR expression using the fluorescent reporter mCherry during intracellular growth of M. tuberculosis. The author also assessed the immunogenicity of SufR by direct ex vivo stimulation of whole blood samples from newly diagnosed and untreated active TB and QFN positive and QFN negative participants. The following queries may be considered before submitting the revision:

Line 249-250. Authors claim a lower bacterial load was observed at higher MOI and reasoned this might be due to the death of THP-1 cells. A cell viability experiment of THP-1 cells in control and infected should be done for the claim.

The authors did not mention the source of the recombinant protein SufR. Did the endotoxin contamination in the recombinant protein get checked before treatment? Moreover, the stability and quality of the purified recombinant protein is not mentioned.

SufR transcript levels do not match with the mCherry reporter gene. Besides the lower stability of mCherry, there may be other epigenetic factors involved in SufR transcription. The author may knock-in the mCherry reporter gene under the SufR promotor in the genome. This may give a better idea of changes in SufR transcript levels.

Line 274-275 There is not enough evidence to claim sufR expression is induced in a subset of bacteria within macrophages and that this population increases over the course of a 72-hr infection. The author could do a western blot to check the changing expression of SufR in the growing population.

Type 2 immune response cytokines were not analyzed during the recombinant SufR protein treatment.

The study has a very small number of samples to compare and draw any conclusions.

The conclusions and data of the manuscript do not add significant information or knowledge to what was previously known about SufR.

Reviewer #4: Overall, it is an interesting study.

Following are my comments or suggestions:

The number of patients in the three groups: Newly diagnosed untreated TB, latently infected (IGRA positive) and healthy uninfected individuals (IGRA negative) are not mentioned.

In the results, it has been mentioned, “sufR expression is induced in a subset of bacteria within macrophages, and this population increases over the course of a 72-hr infection”. This should be mentioned in the conclusion also.

In the results, there is a paragraph titled, “CD8+ T cell phenotypes are poorly stimulated by SufR, but cytokine-secreting B cells are increased”. But under that paragraph there is no data about increased cytokine-secreting B cells. However, in the discussion, it has been mentioned, B cells shows higher frequency of CD19+IL-2+ (not significant) in the QFN pos group (p = 0.07) when compared to active TB following 12-hr SufR stimulation. So, the cytokine-secreting B cells were increased in the QFN pos group rather than active TB. Please clarify and make necessary changes in the manuscript.

In the discussion, the possible reasons for poor cytokine response despite increased sufR expression can be discussed.

The conclusion is sketchy. It can be elaborated more by mentioning all the key findings of the study.

6. PLOS authors have the option to publish the peer review history of their article (what does this mean?). If published, this will include your full peer review and any attached files.

Reviewer #1: **Yes: **Layth Al-Salihi

Reviewer #2: **Yes: **KALAIARASAN ELLAPPAN

Reviewer #3: **Yes: **Seyed Ehtesham Hasain

Reviewer #4: **Yes: **Dr. Noyal Mariya Joseph

---

## [Author Response · Author response to Decision Letter 0]

16 May 2023

Reviewer #2: 

1.Kindly follow the similar version while using gene names eg., mcherry and sufR throughout the manuscript

Lines 95, 98, 230, 248, 360: mCherry was changed mCherry

Line 226: SufR was changed sufR. Where reference is to the protein, “SufR” was retained.

2. The conclusion is missing in this study and highlights the novelty in it.

A conclusion was added, lines 436-439.

3. Kindly explain the demographic profiles of study participants, sample size and calculation.

This information has been included in the supplementary file in Table S4. See lines 272-273 

Line 158-159: Sample sizes were indicated. 

4. Highlight the concentration of cells (eg., per ml) used for FACS study.

For infection studies, THP-1 cells were seeded at 2 x 105 cells/well. Bacterial enumerated at time zero showed an average percentage uptake of ~ 50 % (Figure 2A). For flow cytometry analysis 30 000 events were recorded. 

For the lymphocyte assays, 200 µl cells were run on the flow cytometer and between 105 and 106 events were recorded. These details have been highlighted in the text in lines 120, 149 and 213.

5. Try to highlight the references in discussion section and not in the results (line 211 to 212)

We acknowledge the reviewer’s comment but feel that the reference is necessary to contextualise the study.

Reviewer #3: 

The following queries may be considered before submitting the revision:

Line 249-250. Authors claim a lower bacterial load was observed at higher MOI and reasoned this might be due to the death of THP-1 cells. A cell viability experiment of THP-1 cells in control and infected should be done for the claim.

An increase in cell death due to increased MOI is a well-known phenomenon. We have added a reference, Bettencourt et al 2017 (line 368-369), which showed that an increase in multiplicity of infection and time leads to a decrease in host cell viability, to support this claim.

The authors did not mention the source of the recombinant protein SufR. Did the endotoxin contamination in the recombinant protein get checked before treatment? Moreover, the stability and quality of the purified recombinant protein is not mentioned.

 We acknowledge that information about the purification of the protein should have been included. This information has now been added in lines 166-172. “Recombinant SufR was produced and purified as detailed in Willemse et. al. (2018). Briefly, 6xHis-tagged SufR was expressed in the E. coli Arctic express DE3 strain and purified by two rounds of Ni-IMAC affinity chromatography. Prior to the second HiTrap column, the recombinant protein was concentrated by ultra-filtration, dialysed to remove imidazole and the 6xHis-tag removed by tabaco etch virus protease cleavage. The protein was again concentrated by ultrafiltration and further purified by gel filtration chromatography.”

This affinity chromatography procedure has been shown to be effective in depleting endotoxins (DOI: 10.1016/j.ab.2014.08.020). In addition, while the protein was not specifically checked for endotoxin contamination, endotoxin contamination produces an exaggerated early IL-10 response (10.1016/j.jpedsurg.2004.02.009 ; DOI: 10.1186/1742-2094-9-3). This would have been evident for the intracellular cytokine response (flow cytometry) and for the IL-10 levels in supernatants. 

SufR transcript levels do not match with the mCherry reporter gene. Besides the lower stability of mCherry, there may be other epigenetic factors involved in SufR transcription. The author may knock-in the mCherry reporter gene under the SufR promotor in the genome. This may give a better idea of changes in SufR transcript levels.

We acknowledge that integrating the reporter elsewhere in the genome may influence the transcript levels. However, generating a knock-in may be problematic given the regulatory elements present in this region. The promoter being investigated is upstream of sufR and controls the expression of the entire suf operon (sufR-sufB-sufD-sufC-csd-nifU-sufT). A second promoter is located within sufR and controls the expression of sufB-sufD-sufC-csd-nifU-sufT. Altering the region within sufR was shown to decrease expression of sufB-sufD-sufC-csd-nifU-sufT (DOI: 10.1016/j.redox.2021.102062). Therefore, altering the region by knocking in mCherry may impact expression of sufB-sufD-sufC-csd-nifU-sufT, which in turn could alter expression from the sufR protmoter (due to the feedback mechanism to meet FeS demand). 

Line 274-275 There is not enough evidence to claim sufR expression is induced in a subset of bacteria within macrophages and that this population increases over the course of a 72-hr infection. The author could do a western blot to check the changing expression of SufR in the growing population.

We acknowledge that measuring protein levels would be ideal. We attempted to generate and use of a custom antibody raised against purified recombinant SufR for this purpose. However, this antibody was unable to detect SufR protein in M. tuberculosis H37Rv protein lysates due to low sensitivity (Figure 1). 

Type 2 immune response cytokines were not analyzed during the recombinant SufR protein treatment.

We acknowledge that our cytokine selection is a limitation, and we would have like to include Th2, Th17 and other immunological markers. However, based on our own knowledge and previous literature and due to financial constraints, we decided to evaluate only the selected markers which we identified from literature as immunological markers that have the potential to discriminate between, active TB patients, LTBI individuals and uninfected individuals. 

The study has a very small number of samples to compare and draw any conclusions.

We acknowledge the small sample size as a limitation, future studies will consist of a larger sample size.

The conclusions and data of the manuscript do not add significant information or knowledge to what was previously known about SufR.

We acknowledge the reviewer’s comment, however the work aimed to provide preliminary information about the host immune response to this antigen. 

Reviewer #4: Overall, it is an interesting study.

Following are my comments or suggestions:

The number of patients in the three groups: Newly diagnosed untreated TB, latently infected (IGRA positive) and healthy uninfected individuals (IGRA negative) are not mentioned.

The sample size of the three groups were added to the text in lines 158-159. 

In the results, it has been mentioned, “sufR expression is induced in a subset of bacteria within macrophages, and this population increases over the course of a 72-hr infection”. This should be mentioned in the conclusion also.

We have included the following “sufR expression is induced in a subset of bacteria within macrophages, and this population increases over the course of a 72-hr infection” in the conclusion as suggested. lines 436-439.

In the results, there is a paragraph titled, “CD8+ T cell phenotypes are poorly stimulated by SufR, but cytokine-secreting B cells are increased”. But under that paragraph there is no data about increased cytokine-secreting B cells. However, in the discussion, it has been mentioned, B cells shows higher frequency of CD19+IL-2+ (not significant) in the QFN pos group (p = 0.07) when compared to active TB following 12-hr SufR stimulation. So, the cytokine-secreting B cells were increased in the QFN pos group rather than active TB. Please clarify and make necessary changes in the manuscript

We acknowledge this omission. The text in the results have been updated (see lines 327 – 333) and the data has been included in figure S3 and table S6 in the supplementary file.

In the discussion, the possible reasons for poor cytokine response despite increased sufR expression can be discussed.

This has now been discussed in lines 427 -434.

The conclusion is sketchy. It can be elaborated more by mentioning all the key findings of the study.

A conclusion has been added in lines 436 – 439.

---

## [Decision Letter · Decision Letter 1]

29 May 2023

Investigating Mycobacterium tuberculosis sufR (rv1460) in vitro and ex vivo expression and immunogenicity

PONE-D-23-04734R1

Dear Dr. Monique Joy Williams,

We’re pleased to inform you that your manuscript has been judged scientifically suitable for publication and will be formally accepted for publication once it meets all outstanding technical requirements.

Kind regards,

Wenping Gong, Ph.D.

Academic Editor

PLOS ONE

Additional Editor Comments (optional):

Reviewers' comments:

Reviewer's Responses to Questions

**Comments to the Author**

1. If the authors have adequately addressed your comments raised in a previous round of review and you feel that this manuscript is now acceptable for publication, you may indicate that here to bypass the “Comments to the Author” section, enter your conflict of interest statement in the “Confidential to Editor” section, and submit your "Accept" recommendation.

Reviewer #2: All comments have been addressed

Reviewer #3: All comments have been addressed

2. Is the manuscript technically sound, and do the data support the conclusions?

Reviewer #2: Yes

Reviewer #3: Yes

3. Has the statistical analysis been performed appropriately and rigorously? 

Reviewer #2: Yes

Reviewer #3: Yes

4. Have the authors made all data underlying the findings in their manuscript fully available?

Reviewer #2: Yes

Reviewer #3: Yes

5. Is the manuscript presented in an intelligible fashion and written in standard English?

Reviewer #2: Yes

Reviewer #3: Yes

6. Review Comments to the Author

Reviewer #2: All the comments have been adequately addressed and author’s responses were satisfactory

I have no further comments

Reviewer #3: I have gone through the revised manuscript and also the author's response to the comments of the reviewers. To address the comments of the reviewers, conclusion was added in the lines 436-439. Authors have also added the demographic profiles of study participants, sample size and calculation in the supplementary file in Table S4. Authors have also added the references at a few places in the revised manuscript. The text in the results section has been updated by the Authors in the lines 327 – 333 and the data has been included in figure S3 and table S6 in the supplementary file. In my view, authors have satisfactorily addressed all the queries of the reviewers and revised the manuscript satisfactorily. I recommend this manuscript for publication.

7. PLOS authors have the option to publish the peer review history of their article (what does this mean?). If published, this will include your full peer review and any attached files.

Reviewer #2: **Yes: **Ellappan Kalaiarasan

Reviewer #3: No

---

## [Editor Report · Acceptance letter]

6 Jun 2023

PONE-D-23-04734R1 

Investigating *Mycobacterium tuberculosis sufR (rv1460) in vitro* and *ex vivo* expression and immunogenicity 

Dear Dr. Williams:

I'm pleased to inform you that your manuscript has been deemed suitable for publication in PLOS ONE. Congratulations! Your manuscript is now with our production department. 

Kind regards, 

on behalf of

Dr. Wenping Gong 

Academic Editor

PLOS ONE